# Study of Mechanical Properties and Durability of Alkali-Activated Coal Gangue-Slag Concrete

**DOI:** 10.3390/ma13235576

**Published:** 2020-12-07

**Authors:** Hongguang Zhu, Sen Yang, Weijian Li, Zonghui Li, Jingchong Fan, Zhengyan Shen

**Affiliations:** School of Mechanics & Civil Engineering, China University of Mining and Technology (Beijing), Beijing 100083, China; zhuhgcumtb@163.com (H.Z.); liweijian199407@163.com (W.L.); lizonghuicumtb@163.com (Z.L.); fjccumtb@163.com (J.F.); zyshen1995@163.com (Z.S.)

**Keywords:** alkali-activated coal gangue-slag, chlorine salt corrosion, freezing and thawing cycles, chloride permeability, compressive strength, damage mechanics models

## Abstract

Herein, a new geopolymer is recognized as a potential alternative cementing material of ordinary Portland cement (OPC), which is used for reducing carbon emissions and efficiently recycling the waste. Therefore this paper mainly studied the alkali-activated coal gangue-slag concrete (ACSC) was prepared by using the coal gangue-slag and Na_2_SiO_3_ and NaOH complex activator. The ratio of coal gangue (calcined and uncalcined) coarse aggregate replacing the gravel was 0%, 30%, 50%, 70%, and 100%. The water and salt freeze-thaw resistance, compressive strength, chloride permeation, microstructure, performance mechanism, inner freeze-thaw damage distribution, and mechanics models of ACSC were investigated. Results show that ACSC displayed excellent early age compressive strength, and the compact degree and uniformity of structure were better compared with the ordinary Portland cement (OPC) when the coal gangue replacement rate was less than 50%. The ACSC demonstrated the best chloride penetration resistance under 30% uncalcined coal gangue content, which was less than 27.75% lower than that of using OPC. At the same number cycles, especially in the salt freezing, the calcined coal gangue had lowered advantages of improving resistance freeze-thaw damage resistance. Water and salt accumulative freeze-thaw damage mechanics models of ACSC were established by using the relative dynamic elasticity modulus. The exponential function model was superior to the power function model with better precision and relativity, and the models accurately reflected the freeze-thaw damage effect.

## 1. Introduction

The alkali activation of waste materials (e.g., metakaolin, coal gangue, fly ash, or slag) has become an important area in the development of new green cements (inexpensive and environment-friendly) [1] and can be formulated by wastes and activator. Under extremely high OH^−^ concentrations, Si–O–Si– and –Si–O–Al– vitreous body structure is rapidly dissolved into solution to form [SiO4]^4−^ and [AlO4]^5−^ tetrahedral units. Meanwhile, new–O–Si–O–Al–O– binding materials with three-dimensional network structure are obtained by shrinking and polymerization reaction [2,3]. These materials have high early compressive strength and good resistance to acid, alkali, and freeze–thawing, and can reduce the use of ordinary Portland cement (OPC) and energy consumption. Therefore, these construction materials with a low-carbon footprint have received significant attention worldwide [1,4].

Coal gangue, which has a similar composition to clay [5], is a solid waste produced during coal mining and processing [6,7,8,9,10], and its annual production accounts for about 10% to 15% of the coal output in that year [6,7,11]. Fresh undisturbed coal gangue does not have pozzolanic activity [9], which in general leads to the low strength of coal gangue aggregate concrete and severely limits its scope of use. However, high-temperature calcined coal gangue can prevent the polymerization of silicon tetrahedrons and aluminum tetrahedrons into long chains, and there are several breaking points that form thermodynamically unstable structures so that the pozzolanic activity (SiO_2_, Al_2_O_3_) can be improved. In addition, secondary hydration with Ca(OH)_2_ can influence the mechanical properties and microstructure of concrete [6,12]. Zhang et al. [13] demonstrated that the ground coal gangue hardly reacts with water and has weak cementitious characteristics, but it displays extremely strong cementitious characteristics in alkaline solutions. Multiple scholars have suggested new ideas for the ultra-efficient utilization of coal gangue, which is important for a sustainable and friendly environment and has economic benefits in the research on coal gangue concrete. Hence, it is essential to accelerate the comprehensive utilization of coal gangue resources [14,15].

Alkali activation can actually be represented via two different models, both corresponding to two extremely different conditions in terms of the starting situation and chemical composition [3]. The first model is established by the activation of ground blast furnace slag with a mildly alkaline solution, and the chemical composition was mostly CaO–SiO_2_. The second model of alkali activation has been studied for metakaolin, fly ash, and coal gangue and their main chemical composition is SiO_2_-Al_2_O_3_. A well-known example is a slag with high calcium content that is often used as a raw material to prepare cementitious materials via alkaline activation. Several domestic and international scholars have studied the chemical composition of alkali-activated cementitious materials [16], hydration products and mechanisms [17], types of alkali activators [18], mechanical properties of hydraulically hardened cement [19], alkali-aggregate reaction [20], and durability of these materials [21]. Ranjbar [22] discussed the factors that affect the setting time of fresh geopolymer and discussed the underlying chemical and physical-mechanisms of each parameter. However, there are limited studies on the preparation of cementitious materials by alkali-activated coal gangue [13]. Ma et al. [23] used NaOH and Na_2_SiO_3_ alkali activators to produce coal gangue-slag geopolymer. The results show that the liquid–solid ratio is the main factor that affects the fluidity and strength of alkali-activated coal gangue-slag cementitious materials (AACGS). In addition, Ca^2+^ in the high-calcium slag promotes the exchange with Na^+^, and the product is converted from N–A–S–H gel to C–(A)–S–H gel while the Ca/Si ratio increases. The usage of slag instead of coal gangue can greatly improve the reaction process and increase the strength of AACGS materials. By employing spontaneously combusted or calcined coal gangue, slag, fly ash, and recycled aggregates as raw materials, the recycled concrete of geopolymer can be produced. Liu et al. [24] concluded that the key factors that affect the mechanical properties of recycled concrete are the internal pore structure, micro-cracks, and interface-bonding ability between geopolymer and recycled coarse aggregates.

Fu et al. [2] studied the freeze-thaw resistance of alkali-activated slag concrete, whose experimental results show that ASC has excellent freeze-thaw resistance with a frost-resisting grade of F300 at lowest. Zuo et al. [25] found that the chloride diffusion coefficient of alkali-activated fly ash decreases with the increase of slag content. Ismail et al. [26] found that the chloride diffusion coefficient of alkali-activated slag-fly ash concrete was 0.35 × 10^−12^ m^2^/s when fly ash up to 50 wt % increases the chloride diffusion coefficient. There is a significant reduction of the chloride diffusion coefficient for 75 wt % fly ash mortars where the activator concentration is higher (12% rather than 8%) than in specimens formulated with reduced contents of fly ash.

There are some reports on the chloride ion penetration [25,26], carbonization, and sulfate corrosion of alkali-activated cementitious materials [4], but there is limited research on the double damage of chloride ion corrosion and freeze-thaw damage of alkali-activated coal gangue cementitious materials. Calcined coal gangue has better activity when it reaches 700 °C [5,6], and then the alkali-activated coal gangue-slag can be employed as a cement. Therefore, this test sets the temperature for calcined coal gangue at 700 °C and the ratio of replacing gangue (calcined and uncalcined) with gravel at, respectively, 0%, 30%, 50%, 70%, and 100% to study the chloride permeability. The 28 d and 90 d compressive strength, 90 d chloride iron corrosion compressive strength, and the change in dynamic elastic modulus and mass-loss rate under the conditions of water freezing and salt freezing were obtained. Herein, the water and salt accumulative freeze-thaw damage mechanics models of alkali-activated coal gangue-slag concrete (ACSC) were established by using relative dynamic elasticity modulus and coal gangue replacement rate as a variable, and the microscopic morphology of coal gangue concrete was analyzed under a scanning electron microscope (SEM). This research work can provide a theoretical basis and reference value for ACSC in engineering applications.

## 2. Materials and Methodology

### 2.1. Materials

The slag was classified as Class S 95, and its specific surface area was 450 m^2^/kg, density was 2.96 g/cm^3^, the mass fraction of SO_3_ was 2.0%, the mass fraction of chloride ion was 0.06%, and vitreous content was 85%. The coal gangue was produced by the Tangshan mining area was tested for non-spontaneous combustion. The coal gangue powder was calcined in a muff furnace at a constant temperature of 700 °C for 2 h. The main chemical compositions of precursors were analyzed by X-ray fluorescence spectrometry (XRF) (Table 1). The median sizes d 50 of the coal gangue and slag are 17.365 µm and 10.532 µm, separately.

In the production process of coal gangue aggregate, the coal gangue was crushed by a jaw crusher, ground by a small ball mill and calcined in a muffle furnace at 700 °C for 2 h (Figure 1).

Sand with a fineness modulus of 2.7 was used as fine aggregate, which had an apparent density of 2610 kg/m^3^ and a loose bulk density of 1400 kg/m^3^. Coarse aggregates used were gravel and coal gangue (calcined and uncalcined) with a diameter of 4.75–20 mm. The physical characteristics of the coarse aggregates are shown in Table 2.

Na_2_SiO_3_ and NaOH complex solutions were employed as activators, and the silica modulus *n* (molar SiO_2_/Na_2_O ratio) was 1.3.

### 2.2. Mix Proportion and Specimen Preparation

The mix proportion was tested according to JGJ 12–2006 “Technical specification for lightweight aggregate concrete structures” [27], and trial preparation was performed in the laboratory in the early stage. The mix proportion ACSC ratio is presented in Table 3. W and S represent uncalcined and calcined coal gangue coarse aggregates, respectively, and W, S (0, 30, 50, 70, and 100), respectively represent the same quality instead of gravel 0%, 30%, 50%, 70%, and 100%. The reference specimens were produced using OPC, mix proportion of water–binder ratio was 0.4.

Based on the ratio in Table 3 and combined with GB/T 50082-2009 “Standard for test methods of long-term performance and durability of ordinary concrete” [28], fresh concrete of standard cube specimens (100 × 100 × 100 mm) size were made for freeze-thaw cycles test and compressive strength. After 24 h, the samples were demolded and cured under controlled regimes (20 ± 2 °C, RH ≥ 95%).

### 2.3. Methods

#### 2.3.1. Compressive Strength

According to GB/T 50081-2019 “Standard for test method of mechanical properties on ordinary concrete” [29], the compressive strengths of ACSC after 28 d and 90 d standard curing were determined. Additionally, to study the effect of chloride ion erosion on the compressive strength of concrete, after curing in the standard curing room for 28 d, concrete specimens were immersed in 5 wt/% NaCl solution for 90 d; the solution was changed every 30 d.

According to the standard [29], the compressive strength test adopted load control with 0.5 MPa/s loading speed. The static load channel of a Multichannel dynamic and static fatigue testing machine FLPL204 (Fule Instrument Corp Fuel Instruments & Engineers Pvt. Ltd. Maharashtra, India) was used.

#### 2.3.2. RCM Tests

The test for rapid chloride ions migration coefficient (RCM) of concrete samples was conducted according to the standard GB/T50082-2009 [28]. The chlorides diffusion coefficient *D_RCM_* is calculated according to Equation (1):(1)$$D_{RCM}=2.872\times 10^{-6}\frac{Th\left(X_{d}-3.338\times 10^{-3}\sqrt{Th}\sqrt{X_{d}}\right)}{t}$$

In which, *D_RCM_* is the chlorides diffusion coefficient determined by RCM method (m^2^/s); *T* is the temperature (K); *h* is the height of the specimen (m); *X_d_* is the chlorides diffusion average depth (mm); *t* is the current test time (s).

#### 2.3.3. Freeze-Thaw Cycles

According to the standard [27], the rapid freeze-thaw method was implemented. The test of freeze-thaw cycles was conducted in two modes: water freezing and 5 wt % NaCl solution (mass fraction). The specimens were saturated 4 d before the test. The surface moisture was wiped before placing in the freeze-thaw machine, and the initial mass and dynamic elastic modulus were measured. The freeze-thaw cycle parameters were set in the freeze-thaw testing machine and the time for each cycle to 4 h. The minimum temperature at the center of the test specimen was −18 ± 2 °C, while the highest value was 5 ± 2 °C. The temperature at the center of the test specimen rose from −18 °C to 5 °C in about 1.5 h. The test specimen was swapped up and down every 25 cycles to reduce the test error caused by the temperature difference between the upper and lower surfaces of the prism. When the loss rate of the test specimen reached 5%, and the relative dynamic elastic modulus of the test specimen dropped to 60%, the frost-resistance test was stopped.

#### 2.3.4. Microstructural Characterization of ACSC

Scanning electron microscope (SEM) analysis was employed for ACSC specimens to observe the adhesive interface morphology of aggregate and cement paste after freeze-thaw cycles and chloride ion erosion.

## 3. Results and Discussion

### 3.1. Compressive Strength Analysis

#### 3.1.1. 28 d and 90 d Compressive Strength

As presented in Figure 2a, the 28 d and 90 d calcined or uncalcined gangue concrete compressive strength had a similar development trend. In this paper, the compressive strength of 90 d is lower than that of 28 d is based on the premise of high content of gangue (>50%), while the low content shows better mechanical properties. In this paper, the compressive strength of 90 d is lower than that of 28 d is based on the premise of a high content of gangue (> 50%), while the low content shows better mechanical properties. The compressive strength decreased with the increase in the amount of coal gangue, and the interior looseness and porousness of the gangue led to its low strength. After replacement with gravel, the strength of concrete increased. Except for the W100 group of uncalcined coal gangue group, the 28 d compressive strength of the other groups was greater than that of the OPC. The S30 group had the highest 28 d compressive strength of 58.4 MPa, while the OPC 28 d displayed a 26.7% higher compressive strength. Except for the W70, W100, and S100 groups, the 90 d compressive strength of all other samples was greater than that of the OPC. The S30 group had the highest 90 d compressive strength of 59.3 MPa in the coal gangue concrete, which is 11.5% higher than the 90 d compressive strength of the OPC.

The strength reduction rate was derived from Figure 2b. When the substitution rate was less than 50%, the calcined coal gangue strength decline rate was greater than that of the uncalcined coal gangue. When the substitution rate was greater than 50%, the uncalcined coal gangue strength decline rate was greater than that of calcined coal gangue. At the same mixing amount, the 28 d and 90 d compressive strengths of calcined coal gangue concrete increased as compared with the original coal gangue concrete. This is because the calcined coal gangue produced more active SiO_2_ and Al_2_O_3_, and it could also react with alkali activator to form C–S–H gels, which made the structure denser [30], leading to the calcined coal gangue exhibiting better compressive strength than the uncalcined coal gangue.

By comparing the compressive strengths of concrete at 28 d and 90 d, the 90 d compressive strength significantly increased as compared to the 28 d compressive strength of the OPC. With an increase in the amount of coal gangue, both the uncalcined coal gangue group and the calcined coal gangue group displayed a trend of 90 d compressive strength decrease as compared to the 28 d compressive strength. In the uncalcined group with 70% and 100%, i.e., W70 and W100 groups, and when the mixing amount of calcined coal gangue was 100%, i.e., S100 group, the 90 d compressive strength decreased compared with the 28 d compressive strength. This is because the compressive strength of coal gangue itself was poor, with the internal structure having holes, which caused a decrease in compressive strength of uncalcined coal gangue concrete after mixing the coal gangue coarse aggregates. Therefore, to ensure the stability of the concrete in later stages, the amount of coal gangue should not be more than 50%.

#### 3.1.2. 90 d Chloride Salt Erosion Compressive Strength

Figure 3a represents 90d chloride ion erosion compressive strength under different coal gangue replacement rates, and Figure 3b shows the change rate of 90 d chloride ion erosion compressive strength compared to 90 d compressive strength under different gangue substitution rates (calcined and uncalcined).For the uncalcined coal gangue concrete and calcined coal gangue concrete, the 90 d NaCl corrosion compressive strength decreased with the increase in the mixing amount of coal gangue. For the same mixing amount, the 90 d NaCl erosion compressive strength of the calcined coal gangue corrosion increased compared with that for the uncalcined coal gangue concrete, and the decrease in compressive strength was weakened. The calcined coal gangue polymer concrete than uncalcined coal gangue polymer concrete has better compressive strength.

Comparing the 90 d compressive strength with the NaCl corrosion compressive strength (Figure 3a), both the original coal gangue group and calcined coal gangue group demonstrated that when the mixing amount of coal gangue was 70% or less, the 90 d compressive strength was greater than the 90 d NaCl corrosion compressive strength. As the mixing amount of coal gangue increased, the rate of decrease of 90 d chloride salt corrosion strength slowed down compared to the 90 d compressive strength, and the corrosion compressive strength of 90 d NaCl increased inversely in W100 and S100 samples (Figure 3b).

There is a transition zone between the aggregate and cement stone in OPC, which is not conducive to the mechanical properties and durability of the concrete. However, there is a chemical interaction between the alkali-activated cementitious material and the aggregates, and the aluminum-silicon components participate in the polymerization reaction under alkali activation. Therefore, there is no interface transition zone between the alkali-activated cementitious materials and aggregates [31], and the cement porosity of ACSC cement is smaller than that of the OPC cement paste. In addition, the calcined coal gangue improves the activity of coal gangue, and the crystal phase of the stable internal structure is decomposed. The active SiO_2_ and Al_2_O_3_ react with Ca(OH)_2_ inside the concrete to form a C–S–H gel, which reduces the internal pores of the concrete while the structure becomes dense and strength is improved. These two aspects endow superior compressive strength to ACSC as compared to OPC. As presented in Figure 3, the 90 d chlorine salt corrosion compressive strength WO was 26.25% higher than the 90 d compressive strength of the OPC group. The decrease rate of 90 d chlorine salt corrosion intensity of OPC reached 9.77%, which is the highest. The resisting chlorine salt corrosion of ACSC was superior to OPC. According to Figure 3b, with the increase in coal gangue replacement rate, the corrosion strength slowed down. When the replacement rate was 100%, the 90 d chlorine salt corrosion strength was higher than the 90 d compressive strength. The active Al_2_O_3_ contained in coal gangue reacted with chloride ions to form Friedel’s salt, which was filled in the pore structure and reduced the porosity, decreasing the disadvantages of the NaCl solution immersion, large porosity of the coal gangue coarse aggregate, and strong water absorption capacity. Therefore, when the replacement rate reached 100%, the corrosion intensity increased [32]. However, the longer-period effect of corrosion on strength would be studied in the future.

### 3.2. Chloride Permeation

Chloride corrosion on reinforced concrete is one of the most extensively observed and assessed aspects of concrete durability [26]. Chloride ions arrive from de-icing salt or seawater, which lead to the corrosion of the steel, causing a decrease or even disappearance of the bonding force between the steel bar and concrete, reduction of bearing capacity, and early destruction of the concrete [32,33]. Chloride ions migration coefficient is an important indicator that reflects the durability of concrete, which is primarily related to its permeability. The chloride ion migration coefficient (*D*_RCM_) of concrete was obtained by the rapid chloride ion migration coefficient (RCM) method, wherein the chloride ion permeability of concrete was determined.

Figure 4 shows the *D*_RCM_ of ACSC on the 28 d for different coal gangue replacement rates. As the content of coal gangue increased, the *D*_RCM_ was gradually increased. This is because of the inside loose and porous state of the coal gangue, which led to its weak resistance to chloride ion penetration and diffusion. For the same replacement rate, the *D*_RCM_ of calcined coal gangue was faster than that of the original coal gangue. When the substitution rates were 30%, 50%, 70%, and 100%, the *D*_RCM_ increased by 8.98%, 10.89%, 37.72%, and 23.23%, respectively, as compared to the uncalcined form. During the calcination process of coal gangue, the volatile, free water and structural water is lost, the layered structure and crystal structure are destroyed, the structure becomes loose and porous, internal bond breakage increases and the resistance to chloride ion penetration is decreased [8]; therefore, the gangue after calcination has worse resistance to chloride ion as compared to uncalcined form. Calcined coal gangue has no clear advantages in improving the resistance to permeability of chloride ions.

The impermeability of W0, W30, W50, W70, S30 and S50 groups are better than that of OPC. The permeability of concrete depends on the critical pore size [34], and the pores below 100 nm have the best resistance to chloride ion penetration. Therefore, alkali-activated coal gangue slag cement can modify the pore structure (i.e., porosity, pore size distribution, and interface transition zone) and chemical composition of the pore solution and inhibit the diffusion of chloride ions in the hardened cement paste, mortar, and concrete [35,36] The comparison between the two groups of W0 and OPC specimens shows that the ACSC cementitious material concrete has better resistance to chloride-ion penetration than ordinary silicate concrete, wherein the chloride ion permeability coefficient reduces by 34.28%. The test results are consistent with the reported works [35,36].

### 3.3. Freeze-Thaw Cycles

#### 3.3.1. Water Freeze-Thaw Cycles

##### Mass Loss

Figure 5a,b, represent the mass-loss rates of uncalcined and calcined freezing cycles under different replacement rates of coal gangue, respectively. As shown in Figure 5a,b, with a rise in the number of water freezing-thawing cycles, there was a consistent trend of increase in the mass-loss rate of concrete. At the same content, the mass-loss rate of calcined coal gangue was lower than that of the uncalcined group. When the substitution rates were 30%, 50%, 70%, and 100%, the calcined coal gangue increased, respectively, by 2.54%, −0.13%, 2.02%, and 2.20% as compared to the uncalcined form. When the freeze-thaw cycles were 25 and 50 times, the mass-loss rates of coal gangue concretes were within 1%, micro-cracks appeared on the surface of the test specimens, and slight surface-exfoliation phenomenon occurred in the test specimens in some groups. The average mass-loss rate for specimens with a 100% replacement ratio reached 8.27%. When freeze-thaw cycles were 75 cycles, the mass-loss rates of both types of concretes increased significantly, W100 and S100 had the fastest growth rates at 6.42% and 6.69%, respectively, and surface cracks on the test specimens increased. The freezing pressure was greater than the interface bonding property of the concrete aggregates and cement mortar, and such repetition intensified the surface-exfoliation phenomenon. When the freeze-thaw cycles reached 100 times or more, the mass-loss rate of the two types of concrete continued to increase, and the phenomenon that exfoliation of some specimens was accompanied with missing corners appears.

As the cycles increased, the mass-loss rate of OPC gradually increased, but compared with ACSC, the increase in the mass-loss rate of OPC was relatively gentle. When the water-freezing cycle was 200 cycles, the OPC mass-loss rate was 3.17%, and the WO loss rate was 7.40%. The internal loose coal gangue aggregates clearly increased the mass-loss rate of ACSC, and there were no advantages of mixing the calcined coal gangue with regards to controlling the mass-loss rate; rather, it accelerated the damage of freezing-thawing cycles.

##### Relative Dynamic Elasticity Modulus

Figure 6a,b shows that the relative dynamic elastic modulus changes with the number of freeze-thaw cycles under different replacement rates of coal gangue, respectively. With the increase in the number of water freezing cycles, the relative dynamic modulus of uncalcined and calcined ACSC were decreased. In addition, with more content of coal gangue, there was a more prominent decline in the relative dynamic modulus of calcined coal gangue. At the same content, the dynamic modulus of calcined coal gangue and the uncalcined group had a small difference, and there was no clear advantage of calcined coal gangue. When the replacement rates were 30%, 50%, 70%, and 100%, the modulus of calcined coal gangue was increased by 0%, 0.86%, 9.05%, and 1.17%, respectively, compared to the original coal gangue. When the number of water freezing cycles was 25 and 50 cycles, the relative dynamic modulus of ACSC decreased more gently. At 75 cycles, the relative dynamic modulus of the two types of concrete decreased significantly, of which the relative dynamic modulus of W100 and S100 fall to 67.7% and 65.2%, respectively. When the times of freeze-thaw cycles reached 75 or more, the relative dynamic modulus of the two types of concrete continued to decrease more prominently until the relative dynamic modulus reached 60%, when the test was stopped.

As the times of freeze-thaw cycles increased, the relative dynamic modulus of OPC gradually decreased, and when the water freezing-thawing cycles reached 200 cycles, the relative dynamic modulus decreased to 70.7%. However, the modulus decreased slightly compared to coal gangue concretes, while the relative dynamic modulus of coal gangue geopolymer concrete decreased significantly; hence, coal gangue did not display any clear advantages.

#### 3.3.2. Salt Freezing Cycles

##### Mass Loss

Figure 7a,b, represent the mass-loss rates of uncalcined and calcined salt freeze-thaw cycles under different replacement rates of coal gangue, respectively. Based on Figure 7a,b, the salt freezing cycle damage test shows that with an increase in the number of freeze-thaw cycles, the mass-loss rate of coal gangue geopolymer concrete increased, and with more amount of coal gangue, there was a more prominent increase. For the same content, the mass loss of calcined coal gangue compared with the uncalcined group displayed both upward and downward trends. When the substitution rates were 30%, 50%, 70%, and 100%, the mass loss of calcined coal gangue increased by 8.00%, −10.13%, 9.09%, and 19.44%, respectively, as compared to the uncalcined coal gangue.

When the number of freeze-thaw cycles was 25 cycles, the mass-loss rate of concrete increased slightly, which is similar to that the water freeze-thaw state. Furthermore, the surface of the test specimens had micro-cracks, and the surface of the test specimens of some individual groups displayed exfoliation. When the number of freezing and thawing cycle was 50 cycles, the mass-loss rates of the two types of concrete increased significantly, and the mass-loss rates of W100 and S100 reached 7.2% and 8.6%, respectively. The surface cracks of the test piece increased, damage speed was accelerated, and the damage developed layer-by-layer, forming a loose layer with an uneven surface and the phenomenon of exfoliation intensified. When the freeze-thaw cycles were 75 times or more, the mass-loss rate of the two types of concrete continued to increase, the surface layer of erosion appeared, NaCl crystals aggregated, and crystal pressure was greater than the interface pressure of the material and mortar. In addition, the phenomenon of exfoliation and missing corners of the test specimen indicates that the damage was more severe under the double action of chlorine salt corrosion and freeze-thaw cycles. The mass-loss rates of both calcined and uncalcined coal gangue groups at 125 times salt freezing cycles reached more than 5%, and the mass-loss rate of OPC cement at this time was 4.91%. At 200 salt-freezing cycles, the mass-loss rate of OPC was 8.85%.

As the times of freeze-thaw cycles increased, the mass-loss rate of OPC gradually increased. However, compared with coal gangue geopolymer concrete, the mass-loss rate of OPC at 100 salt–freeze cycles maintained a slow increase. At 100 cycles and above, the overall mass-loss rate of OPC did increase as prominently as that of coal gangue geopolymer concrete. The salt freezing cycles clearly accelerated the mass-loss rate of concrete. The mass-loss rates of W30 and S30 at 125 water freezing cycles were 5.90% and 6.40%, while the value at 75 salt-freezing cycles close to the 125 water-freezing cycles. At 200 salt freezing cycles, the mass-loss rate of W30 and S30 reached 5.51% and 5.99%, and the mass loss of OPC was 2.78 times that of the water freezing cycles.

##### Relative Dynamic Elasticity Modulus

Figure 8a,b shows that the relative dynamic elastic modulus changes with the number of salt freeze-thaw cycles under different replacement rates of coal gangue, respectively. As shown in Figure 8a,b, under the salt freeze-thaw stage, as the number of freeze-thaw cycles increased, the relative dynamic modulus of elasticity of the coal gangue geopolymer concrete decreased. At 50 cycles, the relative dynamic modulus of elasticity of the W100 and S100 group specimens fall below 60%, specifically at 58.3% and 54.3%, respectively.

The relative dynamic modulus of elasticity of OPC cement gradually decreased with the increase in freeze-thaw cycles, but compared with coal gangue geopolymer concrete, the relative dynamic modulus of elasticity of OPC cement was relatively small, while that for coal gangue polymer concrete decreased significantly. At the critical damage stage, salt-freezing damage was more severe than the water–freeze cycles because NaCl reacted with Ca(OH)_2_ in the pores of the concrete to form the expansive double salt Friedel’s salt (3CaO·Al_2_O_3_·CaCl_2_·10H_2_O), which led to delamination. In addition, in combination with chloride ion, NaCl reduced the pH of Ca(OH)_2_ consumption, and the stability of C–S–H gel had equilibrium damage, causing the decomposition of C–S–H gel and accelerating the destruction of concrete [37]. On the other hand, salt freezing increased the saturation of concrete and reduced the freezing point. The pores and capillary channels were filled with free water, and the water absorption, saturation, and water absorption speed increased significantly with the increase in NaCl concentration. Absorbing more water increased the expansion pressure, which led to a rise in the destructive power [38,39].

#### 3.3.3. Freeze-Thaw Damage Mechanics Models of ACSC

The freeze-thaw damage deterioration of concrete was due to the expansion of the frozen volume of the internal structure of concrete at a certain freezing temperature, migration of cold water, and various pressures then caused. Power et al. [40] had proposed the osmotic pressure theory. The saturated vapor pressure of ice and water promotes the migration of cold water to the frozen area. When the pressure exceeds the tensile strength of concrete, the micro cracks are generated, the cracks expand and continue to develop in depth, the strength gradually decreases until the loss of bearing capacity.

The standard parameters of freeze-thaw damage deterioration include the relative dynamic modulus of elasticity, mass-loss rate, and strength attenuation. As mentioned above, it is improper to use the mass loss and strength attenuation model to evaluate the freeze-thaw resistance of ACSC. This is because relative dynamic elasticity modulus or cumulative damage model can expediently test the progress of damage with a spot of samples [2]; thereby, freeze-thaw damage model used relative dynamic elasticity modulus as the damage variable to estimate the freeze-thaw damage degree of ACSC.

##### Freeze-Thaw Cycle Cumulative Damage Model Models

Damage degree of concrete after freeze-thaw cycles can be defined via Equation (2) based on the basal theory of damage mechanics. In other words, relative dynamic elasticity modulus can be used to estimate the damage degree of concrete (damage variable) [2]. Damage degree increases with the reduction of relative dynamic elasticity modulus of concrete after freeze-thaw cycles:(2)$$D_{\left(n\right)}=1-E_{n}E_{0}^{-1}$$

As presented in Equation (2), $E_{0}$ is the dynamic elasticity modulus of concrete without damage, $E_{n}$ is the remnant dynamic elasticity modulus after *n* times freeze-thaw cycles, and $D_{\left(n\right)}$ is the cumulative damage value of concrete after n times freeze-thaw cycles.

According to the above definition, the degree of freeze-thaw cycle cumulative damage is a function of times of freeze-thaw cycles, and the damage models of concrete can be mostly established based on the power function Equation (3) and exponential function Equation (4) (see Figure 9 and Figure 10) [2]. *N* is the times of freeze-thaw cycles; *a*, *b* are the undetermined coefficients in the formula; and e is a constant, which is approximately equal to the 2.71828.
(3)$$D_{\left(n\right)}=1-E_{n}E_{0}^{-1}=aN^{b}$$
(4)$$D_{\left(n\right)}=1-E_{n}E_{0}^{-1}=a\left(1-e^{-bN}\right)$$

Figure 9a,b shows that power function model of freeze-thaw damage of ACSC under different water freezing and salt freezing with different freezing-thawing cycles, respectively. And Figure 10a,b shows that exponential function model of freeze-thaw damage of ACSC under different water freezing and salt freezing with different freezing-thawing cycles, respectively. According to Table 4 and Table 5 and testing results in Figure 9 and Figure 10, the freeze-thaw cycle cumulative damage models based on exponential function models were superior to power function models with better precision.

### 3.4. Compressive Strength and Freeze-Thaw Mechanism

#### 3.4.1. Compressive Strength Mechanism Analysis

Through the analysis of 3.1.1 compressive strength and 3.1.2 chloride salt corrosion compressive strength, it can be concluded that as the coal gangue replacement rate increased, the compressive strength continuously decreased. Figure 11a,b presents the failure modes of coal gangue aggregates concrete and crushed stone aggregates concrete, respectively, (ITZ is the interface transition zone). As per the analysis of the damage mode, the damage in coal gangue concrete was similar to OPC. When the concrete cube test specimen is damaged by compression, the Poisson phenomenon occurs. In other words, the damaged surface extends to the center of the specimen from the upper and lower edges of the test specimen at 45°. The upper and lower sides of the test block were basically intact, and more defects are present closer to the middle part, and the middle part becomes thinner.

Figure 11 also shows that when the coal gangue concrete was damaged by compression, the calcined and uncalcined coal gangue coarse aggregates displayed different degrees of fractures and cracks. The exfoliation and break-off phenomenon of coal gangue aggregates was prominent while the gravel was intact. The gangue aggregates were still covered with cementitious materials, which indicated that the bonding strength of ACSC was higher than that of the coal gangue aggregates but lower than that of gravel. Coal gangue is composed of carbonaceous, muddy, and sandy shale, and its compressive strength is lower than that of gravel. The structure of coal gangue is layered, with micro-cracks inside the loose internal structure. As shown in Figure 12, the part of the interface area of coal gangue aggregate and ACSC had cavities and pores, which can affect the compression resistance of ACSC to a certain extent. As the replacement of gravel with coal gangue increased, the strength improvement advantages of ACSC were dominated by the disadvantages brought by the increase in coal gangue aggregates. Furthermore, when the replacement rate was more than 50%, the compressive strength decline rate accelerated.

To summarize, the coal gangue concrete was damaged by compression, and the compressive strength decreased with the increase in coal gangue replacement rate because the broken gravel ACSC were separated from each other, leading to a decrease in the bonding strength, which then led to the damage of concrete by compression. The strength of the coal gangue coarse aggregates itself is low, the internal structure is relatively loose, and there are cavities in a small part of the area where the coal gangue coarse aggregate is bonded to the ACSC; thereby, reducing the ability of the ACSC to resist compression damage.

Compared with the OPC group, when the coal gangue replacement rate did not exceed 50%, the gangue concrete at 28 d, 90 d, and 90 d chlorine salt corrosion strength were all improved. However, all the compressive strengths were lower than that for the W0 group, and the decrease was of various degrees as a function of coal gangue replacement rate. Figure 13a–d, represent the scanning electron microscope images of OPC group, W0group, W100 group and S100 group, respectively. From Figure 13a, in the OPC, there were needle-like or columnar AFt crystals cement hydration products, network-like C–S–H, and thin plate-like Ca(OH)_2_ crystals [30]. The hydration products were interlaced, and the interstice and voids in the overall structure led to a decrease in the density of the structure. However, Figure 13b shows that the ACSC was more tightly bonded with the gravel. The silicon-aluminum component in the aggregate underwent polymerization reaction under the alkali activation. No interface transition zone occurred in OPC, and the surrounding of the bonding interface was covered with C-S-H gel. There was a large amount of C–S–H gels that made the adhesive interface structure denser, which is a key to improve the mechanical properties at the microscopic level, displaying the difference in the compressive strength at the macroscopic level. The compact degree and uniformity of structure were better compared with the OPC [22].

Comparing Figure 13c,d, the through-calcined coal gangue geopolymer concrete demonstrated better compressive properties than the uncalcined coal gangue group. This is because after the gangue coarse aggregates were calcined, its steady-state surface crystals were cracked and destroyed, and the content of active SiO_2_ and Al_2_O_3_ increased, which endowed the coal gangue coarse aggregates with pozzolanic activity [6]. Meanwhile, some researchers also discovered the same regular, calcined coal gangue pozzolanic activity mainly depended on the degree of non-crystallization kaolinite. Cao [30] resulted show that heating temperature directly affects the mineralogy and crystallinity of calcined coal gangue, kaolinite, in coal gangue transforms into metakaolin, which is an irregular and amorphous phase as calcined at 600 °C ~800 °C because of the dehydroxylation of Al–(O,OH) octahedrons together with the depolymerization of Si-O tetrahedrons, the amorphous metakaolin is of high pozzolanic activity. Dong [6] and Wu [41] have discovered that the active SiO_2_ and Al_2_O_3_ components in the coal gangue, with cement hydration products at a certain degree of secondary hydration reaction, can facilitate a more thorough hydration process and improve the microstructure and macro mechanical properties of cement mortar. Liu [42] showed that pozzolanic reaction contribution to enhanced microstructure (pore and grain size refinement) and hence improved mechanical strength of cementitious material at a later stage. The activated SiO_2_ and Al_2_O_3_ obtained by calcining coal gangue coarse aggregate can also react with the composite alkali-activator to generate C–S–H and other gels [23,24]. Hence, the C–S–H gel made the bonding of calcined coal gangue and cementitious materials denser [23,30,43], and the overall concrete structure became more compact. Therefore, the calcined coal gangue geopolymer concrete displayed better compressive properties than the uncalcined coal gangue groups [21,23].

#### 3.4.2. Freeze-Thaw Mechanism Analysis

In the early stage of freeze-thaw cycles and salt–freeze cycles, the mass-loss rate of coal gangue concrete was relatively small, the relative dynamic modulus of elasticity decreased more slowly. The damage in the early stage of freeze–thawing was concentrated in the interior of the concrete, and the appearance of internal cracks led to a decrease in the relative dynamic modulus of elasticity. The phenomenon only appeared on the surface layer of the test specimens, and the mass-loss rate decreased. As the number of freeze-thaw cycles increased, the internal microcracks continued to expand. Figure 14a,b, represent the scanning electron microscopy of group W100 and S100 in the freeze-thaw cycle for 100 times, respectively. As shown in Figure 14b, the cracks were formed on the bonding interface of ACSC and aggregates and the cementitious material itself, and the phenomenon of denudation and unfilled corner of the test specimen was severe with the exposure of the aggregates; hence, the mass-loss rate increased.

As per the results, the salt-freezing anti-frost properties of the test specimens were worse as compared with water-freezing properties. There is freezable water inside the concrete, which froze and repeatedly expanded during the freeze-thaw process, resulting in the freezing pressure. During the freeze expansion of the frozen water, the binding strength between the aggregates in the concrete was continuously destroyed, which gradually increased the internal micro-cracks in concrete, exacerbating the corrosion degree of the concrete surface. Hence, the anti-frost property gradually decreased until the destruction of the concrete. During the salt–freeze cycles, NaCl reacted with Ca(OH)_2_ in the pores of the concrete to generate the expansive Friedel’s salt (3CaO·Al_2_O_3_·CaCl_2_·10H_2_O), which led to delamination. The combination with chloride ion made Ca(OH)_2_ reduce the consumption of pH. The balance of C–S–H gel stability was destroyed, resulting in the decomposition of C–S–H gel and acceleration of the destruction of coal gangue concrete [43]. On the other hand, salt freezing improved the saturation of concrete and reduced the freezing point. The pores and capillary channels were filled with free-water, and the saturation and absorption rate of capillary water absorption increased significantly with the increase in NaCl concentration. The material absorbed more water, causing a rise in the expansion pressure, which led to an increase in the destructive power [40]. The presence of NaCl significantly increased the internal saturation and growth rate of coal gangue geopolymer concrete, and the expansion rate and freezing pressure produced by the freezing of the freezable water were greater. The internal microcracks were quickly formed, and the cracks gradually converged, which further intensified the freeze-thaw damage degree of coal gangue geopolymer concrete.

Compared with uncalcined coal gangue concrete, the calcined coal gangue concrete demonstrated no clear advantages in anti-frost properties. Owing to calcination, the components of the organic medium in calcined coal gangue coarse aggregates were decomposed and volatilized. Compared with the aggregates, the internal structure of calcined coal gangue coarse aggregates was looser, and there were more pores. The result shows that the critical areas of the calcined coal gangue group were affected by the expansion of ice and became bigger and continued to destroy the micro cracks of coal gangue coarse aggregates. Additionally, in the bonding interface of the coal gangue coarse aggregate and cementitious material, and the coal gangue-slag geopolymer cementitious material, it caused the expansion of cracks and exacerbated the freeze-thaw damage degree of the coal gangue geopolymer concrete, which then weakened the advantages of calcined coal gangue, and hence, the calcined coal gangue group did not display better anti-frost properties than the uncalcined coal gangue group.

#### 3.4.3. Cementing Mechanism

Geopolymer is a novel alkali-activated inorganic polymeric cementitious material used as a building material and has a spatial three-dimensional network bonding structure composed of inorganic [SiO4]^4−^ and [AlO4]^5−^ tetrahedra. In the process of dissolution, hydrolysis, polycondensation, and hardening under alkali-activated conditions, the aluminosilicate raw materials are hydrolyzed to form specific silicon and aluminum hydroxide ions in an alkaline solution. The ions are hydrolyzed to form single [SiO(OH)_3_]^−^ and [Al(OH)_4_]^−^, and have polycondensation reactions with each other to form ionic groups. Then the ionic groups are interwoven to form a network structure, which expands to form a geopolymer material of a certain strength. Some references [3,13] also mention that there are two groups of [SiO4]^4−^ and [AlO4]^5−^ tetrahedra in the geopolymer system. Under the action of OH^−^, the covalent bond of S–O–S and A–O–A is broken in the structure, which gets into the solution as the ion form and combines to form a three-dimensional polymeric aluminosilicate structure. The internal composition has a zeolite-like structure.

Calcined coal gangue and slag are commonly employed geopolymer materials. High-temperature calcined coal gangue prevents the polymerization of silicon tetrahedrons and aluminum oxytrigons from aggregating into long chains. There are several fracture points that form a thermodynamically unstable structure. These points can improve the pozzolanic activity (SiO_2,_ Al_2_O_3_) of coal gangue with their greater solubility. Slag is the most commonly used high-calcium auxiliary cementitious material. CaO polymerizes with [SiO4]^4−^ and [AlO4]^5−^ to form C–A–S–H gel. Zhang et al. [4] also mentioned that in the alkali-activated coal gangue-slag cementitious materials, the hydration products are C–A–S–H gel, C–S–H gel, and N–A–S–H gel. The three hydration products interweave into a disordered network structure, making the concrete structure dense. After solidification, part of the free water exists in the reactants as structured water, and its final product consists of an ionic bond and covalent bond with the supplement of intermolecular forces. However, the traditional cement primarily has intermolecular forces and hydrogen bonding, and there are numerous hydrated crystals and amorphous substances such as AFt and Ca(OH)_2_ coarse crystals in the cement system. The geopolymer has an irregular three-dimensional network structure that is connected with the top corner of the aluminum–silica tetrahedra and the aluminum–oxygen tetrahedron, and an amorphous to semi-crystalline solid material is formed with the network voids filled with alkali metal cations and alkaline earth metal cations.

## 4. Conclusions

Herein, an investigation on the mechanical properties, durability, and related deterioration mechanism of coal gangue geopolymer concrete was performed by considering the content of coal gangue (calcined and uncalcined) coarse aggregate as variables. The research on the mechanical properties primarily included the compressive strength analysis, while the research on durability was primarily the analysis of the resistance chloride penetration and frost resistance. The following conclusions can be drawn based on this study:ACSC displayed excellent early age strength compared with OPC, and the coal gangue can satisfy the compressive strength requirement of concrete aggregates. The development of the compressive strength decreased with the increase in coal gangue content, and 90 d compressive strength (compared with 28 d) reached the decrease when the coal gangue replacement rate was above 50%.The effect of the coal gangue content on the development of 90 d chloride corrosion compressive strength was similar to that on the development of 28 d compressive strength. The 90 d chloride ion corrosion compressive strength with coal gangue content decreased first, and then increased, and increased when the coal gangue content was 100%.The ACSC showed the best chloride penetration resistance under 30% uncalcined coal gangue content, which was less than 27.75% lower than that of using OPC. The *D*_RCM_ values gradually increased with the coal gangue content. At the same content, the growth trend and *D*_RCM_ of calcined coal gangue concrete were higher than the uncalcined form.At the same cycles, more was the content of coal gangue, the speed of salt–freeze cycle was faster than that of freeze–thawing, higher was the mass-loss rate, lower was the relative dynamic modulus of elasticity, damage of calcined coal gangue concrete in each group was more severe, and hence, considering the low-carbon environmental energy consumption, this is not a recommended process to calcined coal gangue.As for the freeze-thaw damage model with coal gangue as a variable freeze-thaw cycle, cumulative damage models based on exponential function models were superior to power function models with better precision, which can provide a reference for the quantificational design of freeze-thaw resistance and microcosmic studies on ACSC after freeze-thaw cycles reaction by mechanics methods.

## Figures and Tables

**Figure 1 materials-13-05576-f001:**
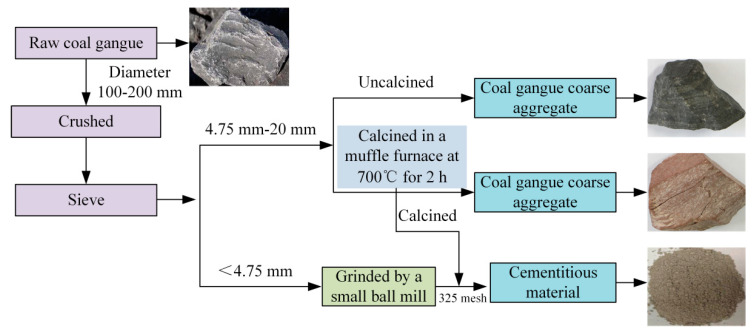
Production process of coal gangue aggregate.

**Figure 2 materials-13-05576-f002:**
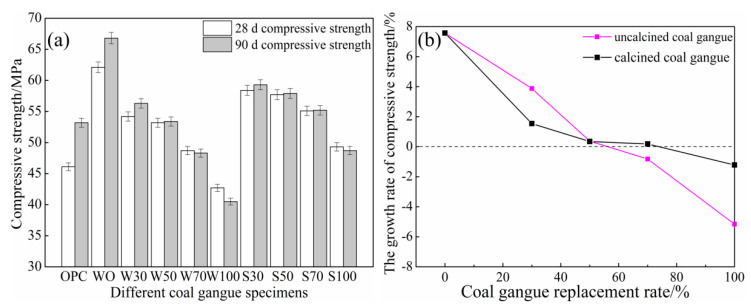
Compressive strength of (**a**) 28 d, 90 d and (**b**) 28 d-90 d growth rate.

**Figure 3 materials-13-05576-f003:**
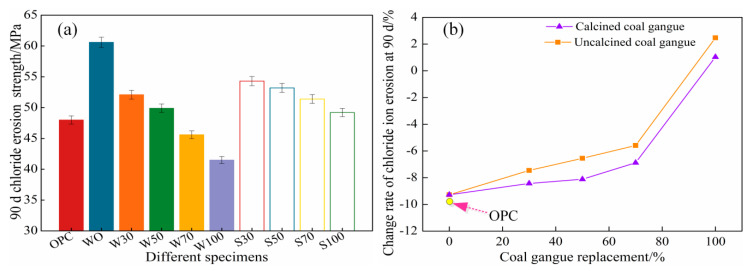
The 90 d chloride ion corrosion compressive strength of (**a**) and (**b**) growth rate.

**Figure 4 materials-13-05576-f004:**
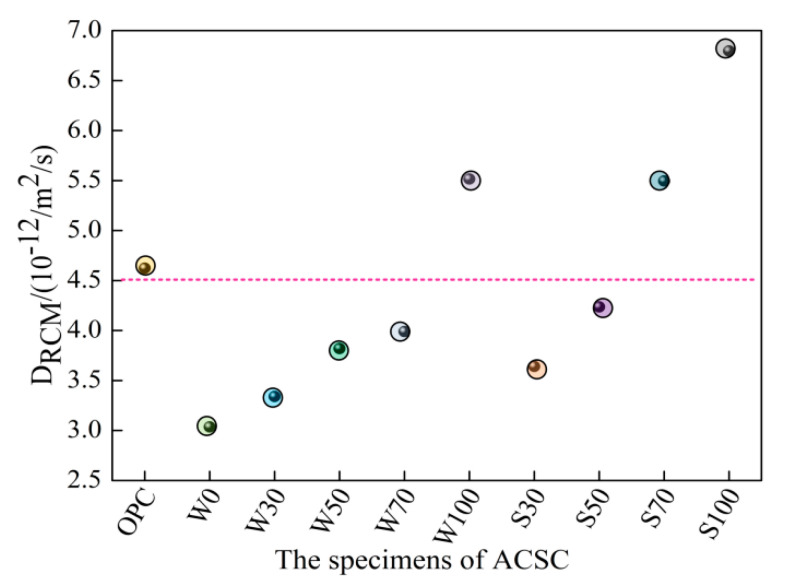
Test method for rapid chloride ions migration coefficient.

**Figure 5 materials-13-05576-f005:**
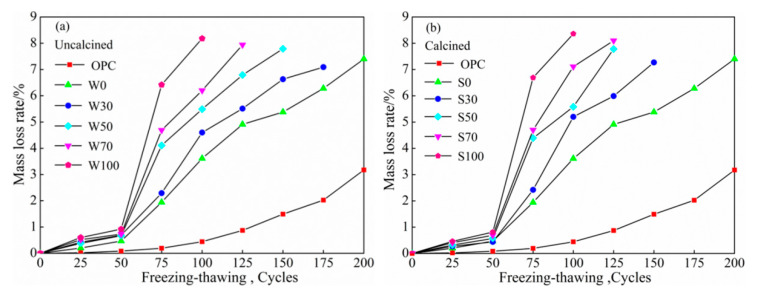
The relationship between the number of freezing-thawing cycles and the mass-loss rate of (**a**) uncalciend and (**b**) calcined.

**Figure 6 materials-13-05576-f006:**
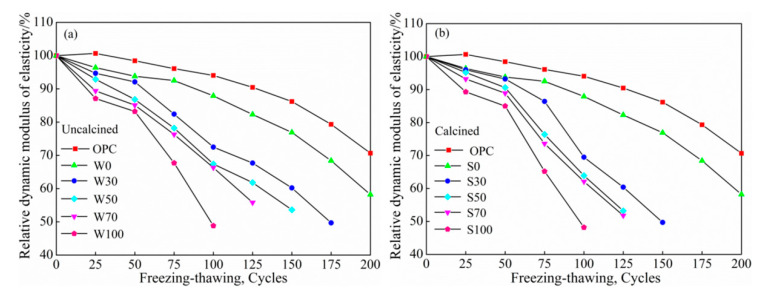
The relationship between the number of freezing-thawing cycles and the relative dynamic modulus of elasticity of (**a**) uncalcined and (**b**) calcined.

**Figure 7 materials-13-05576-f007:**
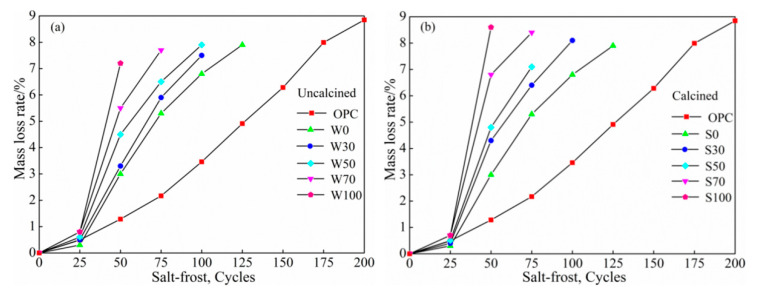
The relationship between cycle times of salt freeze-thaw and mass-loss rate of (**a**) uncalcined and (**b**) calcined.

**Figure 8 materials-13-05576-f008:**
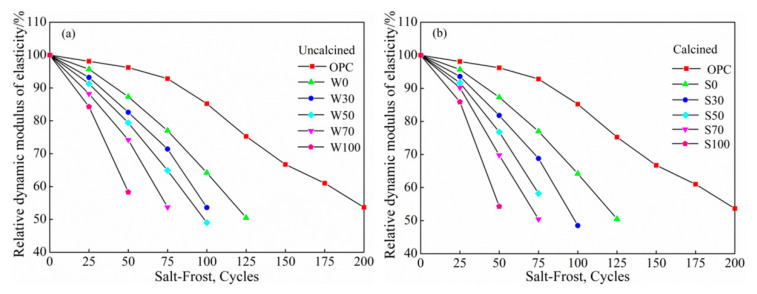
The relationship between the salt-frost and the relative dynamic modulus of elasticity of (**a**) uncalcined and (**b**) calcined.

**Figure 9 materials-13-05576-f009:**
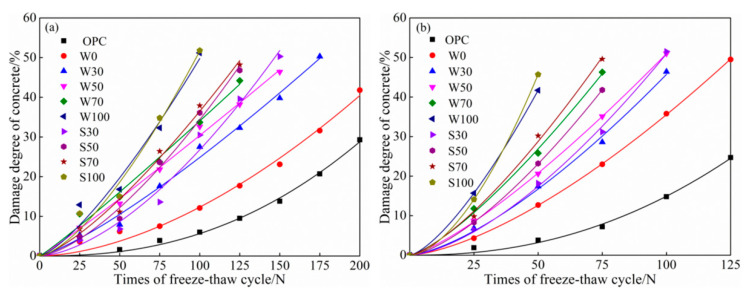
Accumulative freeze-thaw damage power function model of (**a**) water freeze-thaw cycle and (**b**) salt freeze-thaw cycle of alkali-activated coal gangue-slag concrete (ACSC).

**Figure 10 materials-13-05576-f010:**
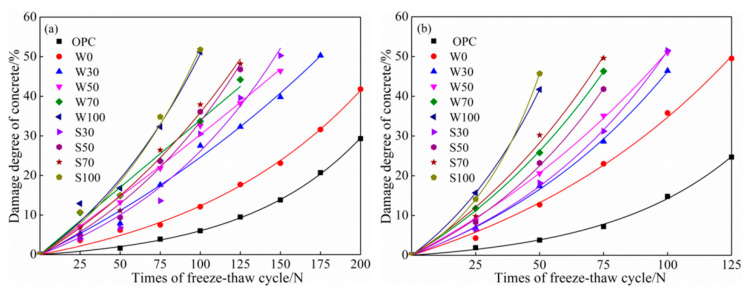
Accumulative freeze-thaw damage exponential function model of (**a**) water freeze-thaw cycle and (**b**) salt freeze-thaw cycle of alkali-activated coal gangue-slag concrete (ACSC).

**Figure 11 materials-13-05576-f011:**
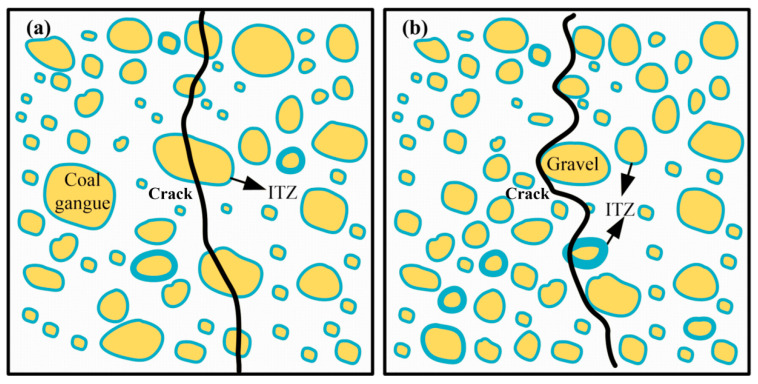
Internal failure pattern of (**a**) coal gangue aggregates and (**b**) crushed stone aggregates of alkali-activated coal gangue-slag concrete (ACSC).

**Figure 12 materials-13-05576-f012:**
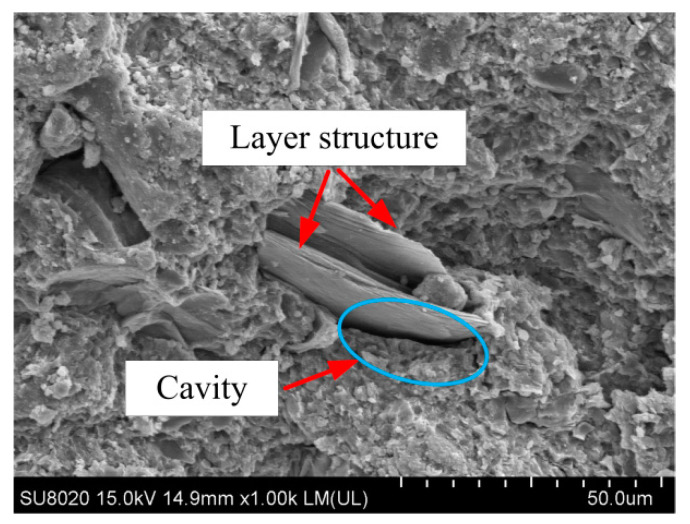
SEM diagram of coal gangue aggregate.

**Figure 13 materials-13-05576-f013:**
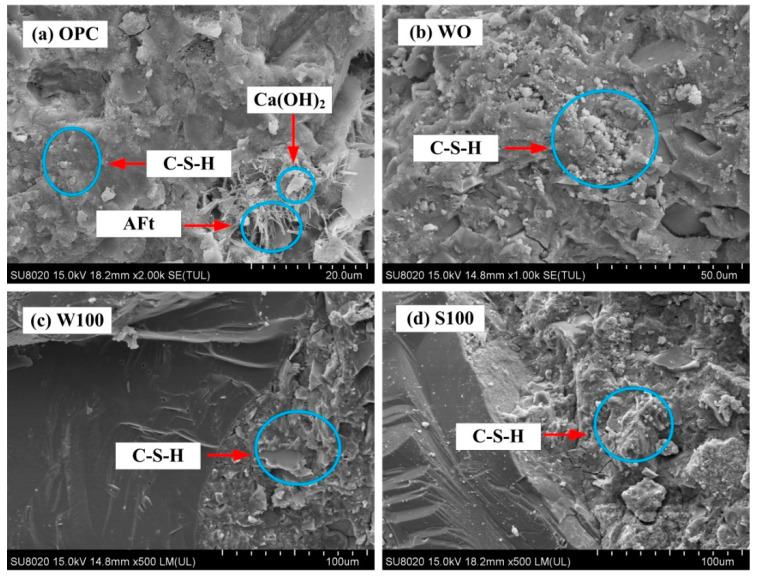
SEM images of the bonding interface between aggregate and cementitious material of (**a**) OPC, (**b**) W0, (**c**) W100 and (**d**) S100.

**Figure 14 materials-13-05576-f014:**
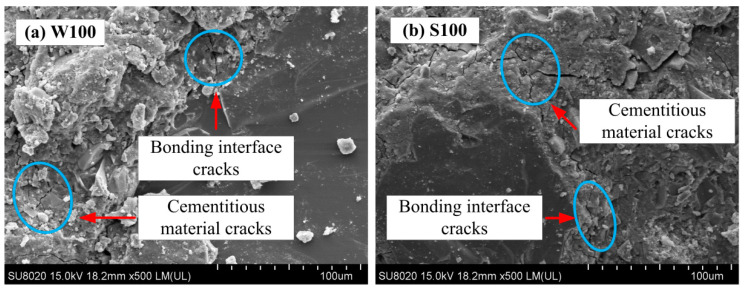
SEM image of ACSC freeze-thaw cycles of (**a**) W100 and (**b**) S100.

**Table 1 materials-13-05576-t001:** Chemical composition of coal gangue and slag (wt %).

Precursor	Al_2_O_3_	SiO_2_	CaO	Fe_2_O_3_	Na_2_O	MgO	TiO_2_	LOI
OPC	4.65	21.18	63.25	3.78	0.18	3.26	1.55	2.15
Slag	14.04	30.58	38.43	0.35	0.57	10.57	1.93	1.17
Coal Gangue	36.78	56.56	0.62	1.95	0.42	0.22	2.10	1.32

**Table 2 materials-13-05576-t002:** Physical characteristics of coarse aggregates.

Types	Particle Size Range /mm	Apparent Density kg/m^3^	Crushing Index Value /%	Water Absorption /%
Ordinary Gravel	4.75–20	2680	16.8	1.51
Calcined Coal Gangue	4.75–20	2605	13.2	6.10
Uncalcined Coal Gangue	4.75–20	2678	13.8	4.09

**Table 3 materials-13-05576-t003:** Mix proportions of alkali-activated coal gangue-slag concrete (kg/m^3^).

Specimens	Cementitious Materials		Water	Na_2_SiO_3_ Solution	NaOH Solution	Aggregates			Compressive Strength/MPa	
	Coal Gangue	Slag				Sand	Coal Gangue	Gravel	28 d	90 d
OPC	-	-	200	-	-	565	-	1050	46.1	53.2
W0	280	280	138.24	172.83	28.02	565	0	1050	62.1	66.8
W30	280	280	138.24	172.83	28.02	565	315	735	54.2	56.3
W50	280	280	138.24	172.83	28.02	565	525	525	53.2	53.4
W70	280	280	138.24	172.83	28.02	565	735	315	48.7	48.3
W100	280	280	138.24	172.83	28.02	565	1050	0	42.7	40.5
S30	280	280	138.24	172.83	28.02	565	315	735	58.4	59.3
S50	280	280	138.24	172.83	28.02	565	525	525	57.7	57.9
S70	280	280	138.24	172.83	28.02	565	735	315	55.1	55.2
S100	280	280	138.24	172.83	28.02	565	1050	0	49.3	48.7

W stands for uncalcined, S stands for calcined.

**Table 4 materials-13-05576-t004:** Salt and water freezing damage power function parameters of ACSC.

Specimens	Freeze-Thaw Damage			Salt-Frost Damage		
	a	b	R^2^	a	b	R^2^
OPC	0.00010	2.35200	0.995	0.00010	2.22476	0.991
W0	0.00454	1.71661	0.984	0.03655	1.49399	1.000
W30	0.08091	1.24357	0.988	0.05778	1.44916	0.994
W50	0.21719	1.07285	0.992	0.13139	1.29431	0.999
W70	0.23727	1.07831	0.977	0.15520	1.31775	0.992
W100	0.10563	1.33660	0.936	–	–	–
S30	0.01485	1.62797	0.977	0.04073	1.54847	0.996
S50	0.03502	1.49583	0.985	0.07563	1.46278	1.000
S70	0.07412	1.34605	0.981	0.13653	1.36781	0.999
S100	0.05865	1.47243	0.958	–	–	–

**Table 5 materials-13-05576-t005:** Accumulative salt and water freezing damage exponential function parameters of ACSC.

Specimens	Freeze-Thaw Damage			Salt-Frost Damage		
	a	b	R^2^	a	b	R^2^
OPC	−2.07457	−0.01361	0.996	−2.29772	−0.01975	0.998
W0	−8.93199	−0.00866	0.996	−21.13862	−0.00972	0.997
W30	−53.44089	−0.00378	0.990	−21.14657	−0.01159	0.999
W50	−231.07696	−0.00123	0.994	−43.44013	−0.00779	0.999
W70	199.58951	0.00002	0.985	−34.08198	−0.01142	0.999
W100	−29.42573	−0.01001	0.977	−23.93107	−0.02018	1.000
S30	−16.35335	−0.00955	0.974	−18.03187	−0.01349	0.999
S50	−21.70524	−0.00935	0.983	−20.09297	−0.01504	0.998
S70	−36.35061	−0.00687	0.984	−35.29314	−0.01178	0.991
S100	−22.60776	−0.01197	0.981	−11.36057	−0.03228	1.000

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
