# Peer review of "Study of Mechanical Properties and Durability of Alkali-Activated Coal Gangue-Slag Concrete"

_materials, 2020, doi:10.3390/ma13235576_

Round 1

Reviewer 1 Report

The topic is interesting but the there is a major comment on the presentation; the current version is a long report with few discussion.  Make the text much shorter with more discussions.

_The abstract should be re-write fully; highlight the objective and outcomes of the study.

_Despite a discussion about the chemical composition of coal gangue has been provided,  discussion about the physical properties of the material was missed. I suggest to compare the physical properties of coal gangue with a similar porous aggregates such as bottom ash and scoria. Some references are by the following:

“Microstructural characterization and mechanical properties of bottom ash mortar”

“Safe disposal of coal bottom ash by solidification and stabilization techniques”

_The effects of thermal activation of the coal gangue need to be elaborated. These effects has already been discussed in literatures, and there might be changes in both amorphousness and physical properties of the material, here are some papers discussed about thermal activation of aluminosilicate sources and their performance in geopolymers.

“Hardening evolution of geopolymers from setting to equilibrium: A review”

“Influence of preheating of fly ash precursors to produce geopolymers”

-Include the difference between Set S and W in the table 3

_Figure 1, separate the specimens in part “a”; the same goes for figure 2.

_What does the pink line shows in  figure 3?

_ Why do you need a calcination of coal gangue if the it reduces the chemical penetration resistance, while does not have influence on mechanical properties?

_The standard deviations for the plots were missed.

_Elaborates the reasons for higher resistance against freeze and thaw cycles in calcined specimens.

_Remove all the hypothetical information on SEMs unless there is an EDS analysis.

_ The English should be revised seriously…e.g.” Compared with OPC, the W0, W30, W50, W70, S30, and S50 groups have better resistance to permeability of chloride ion than that of the OPC”

_Figure-captions are not instructive.

_ 3 models of freeze-thaw damage mechanics models of ACSC are redundant as the predictions are very close, unless you make a justification on the necessity of having all.

Author Response

Dear reviewer:

Many thanks for the comments and suggestions of the paper. The following are the answers and revisions I have made in response to the questions and suggestions. Words in blue are the changes I have made in the text.

  1. The abstract should be re-write fully; highlight the objective and outcomes of the study.

Response: Thank you for your advice and we have revised them.

  1. Despite a discussion about the chemical composition of coal gangue has been provided, discussion about the physical properties of the material was missed. I suggest to compare the physical properties of coal gangue with a similar porous aggregates such as bottom ash and scoria. Some references are by the following:

“Microstructural characterization and mechanical properties of bottom ash mortar”

“Safe disposal of coal bottom ash by solidification and stabilization techniques”

Response: Thank you for your advice and we have revised them.

  1. The effects of thermal activation of the coal gangue need to be elaborated. These effects has already been discussed in literatures, and there might be changes in both amorphousness and physical properties of the material, here are some papers discussed about thermal activation of aluminosilicate sources and their performance in geopolymers.

“Hardening evolution of geopolymers from setting to equilibrium: A review”

“Influence of preheating of fly ash precursors to produce geopolymers”

Response: Thank you for your advice and we have revised them.

  1. Include the difference between Set S and W in the table 3.

Response: Thank you for your advice and we have revised them.

  1. Figure 1, separate the specimens in part “a”; the same goes for figure 2.

Response: Thank you for your advice and we have revised them.

  1. What does the pink line shows in figure 3?

Response: Thank you for your advice and we have revised them. The pink line indicates that the chloride ions migration coefficient is greater than or less than the chloride diffusivity boundary of ordinary Portland concrete.

  1. Why do you need a calcination of coal gangue if the it reduces the chemical penetration resistance, while does not have influence on mechanical properties?

Response: Thank you for your advice. As can be seen from the graph, firstly, the substitution rate of calcined coal gangue is less than 50% and the chloride ion diffusion coefficient is lower than that of ordinary Portland concrete, which shows that the impermeability of calcined coal gangue ground polymer concrete is better than that of ordinary concrete. Secondly, from Fig. 1(a) of the article, we know that the compressive strength of calcined coal gangue concrete is better than that of uncalcined coal gangue. We know from the above conclusion that, calcined coal gangue can improve the compressive strength of concrete, but its impermeability is better than ordinary concrete in a reasonable range(≤50%).

  1. The standard deviations for the plots were missed.

Response: Thank you for your advice and we have revised them.

  1. Elaborates the reasons for higher resistance against freeze and thaw cycles in calcined specimens.

Response: Thank you for your advice and we have revised them. The frost resistance of calcined coal gangue concrete is close to that of uncalcined coal gangue polymer concrete when the freeze-thaw cycles are less than 50 times, but the frost resistance of calcined coal gangue is poor when the cycles are more than 50 times. The main reason is that the layer structure of coal gangue forms micro-cracks on the surface during crushing, the interior is loose and porous, the water absorbing ability is strong, the freezing water can be increased, the icing pressure was increased, the anti-freezing performance is poor. Secondly, calcined coal gangue structure more loose, more pores, resulting in maximum icing pressure. Therefore, the frost resistance of calcined coal gangue is worse than that of uncalcined coal gangue.

  1. Remove all the hypothetical information on SEMs unless there is an EDS analysis.

Response: Thank you for your advice and we have revised them.

  1. The English should be revised seriously…e.g.” Compared with OPC, the W0, W30, W50, W70, S30, and S50 groups have better resistance to permeability of chloride ion than that of the OPC”

Response: Thank you for your advice and we have revised them.

  1. Figure-captions are not instructive.

Response: Thank you for your advice and we have revised them.

  1. 3 models of freeze-thaw damage mechanics models of ACSC are redundant as the predictions are very close, unless you make a justification on the necessity of having all.

Response: Thank you for your advice and we have revised them.

Please check the attachment!

Reviewer 2 Report

It is an interesting article presenting the Mechanical Properties and Durability of Alkali-activated Coal Gangue - Slag Concrete. The authors should address the following contents to improve the quality of the paper:

Line 27: 1. Introduction

Comment 1: One of the major papers on this topic is mentioned below. It is highly related to this subject. I advise the authors to cite it.

“The past and future of sustainable concrete: A critical review and new strategies on cement-based materials”

Line 104: “Table 1. chemical composition of coal gangue and slag (wt.%). 104”

Comment 2: Change “chemical” to “Chemical”

Line 109: Table 2. characteristics of the coarse aggregates. 109

Comment 3: Change “characteristics” to “Characteristics”

Line 115-116: “W and S represent 115 uncalcined and calcined coal gangue coarse aggregates, respectively”

Comment 4: It is confusing for the readers to remember that W is for uncalcined and S represents calcined throughout the whole paper. How about U for uncalcined and C for calcined? Isn’t it easier?

Line 112-122: 2.2. Mix proportion and specimen preparation

Comment 5: The authors did not mention the workability property (Slump) of concrete mixes. Were they the same? What was fixed?

Line 131-132: “According to the standard [26], the compressive strength test adopted load control with 0.5-1.0 131 MPa/s loading speed”

Comment 6: How did the authors accurately controlled this speed? Was the speed manually controlled? If yes, then it is impossible to stay in the range for all the samples.

Author Response

Dear reviewer:

Many thanks for the comments and suggestions of the paper. The following are the answers and revisions I have made in response to the questions and suggestions. Words in blue are the changes I have made in the text.

  1. Comment 1: One of the major papers on this topic is mentioned below. It is highly related to this subject. I advise the authors to cite it.

“The past and future of sustainable concrete: A critical review and new strategies on cement-based materials”

Response: Thank you for your advice and we have revised them. This reference has been cited.

  1. Comment 2: Change “chemical” to “Chemical”

Response: Thank you for your advice and we have revised them.

  1. Comment 3: Change “characteristics” to “Characteristics”

Response: Thank you for your advice and we have revised them.

  1. Comment 4: It is confusing for the readers to remember that W is for uncalcined and S represents calcined throughout the whole paper. How about U for uncalcined and C for calcined? Isn’t it easier?

Response: Thank you very much for your comments, the unified expression of the current research group is calcined (S) and uncalcined (W). I'll consider using your approach in the rest of the presentation.

  1. Comment 5: The authors did not mention the workability property (Slump) of concrete mixes. Were they the same? What was fixed?

Response: Thank you for your advice.

  1. Comment 6: How did the authors accurately controlled this speed? Was the speed manually controlled? If yes, then it is impossible to stay in the range for all the samples.

Response: Thank you for your advice. According to the standard [1], the compressive strength test adopted load control with 0.5 MPa/s loading speed. The static load channel of Multi-channel Dynamic and Static Fatigue Testing Machine FLPL204 (Fule Instrument Corp) was used.

[1] Standards China, GB/T 50081-2019: Standard for test method of mechanical properties on ordinary concrete, Beijing, 2019.

please check the attachment!

Reviewer 3 Report

This paper investigates the effect of the usage of coal gangue-slag as a binder and coarse aggregate on mechanical properties and durability of alkali-activated coal gangue-slag concrete (ACSC). The water and salt freeze-thaw resistance, compressive strength, chloride permeation, microstructure, performance mechanism, inner freeze-thaw damage distribution, and mechanics models of ACSC were investigated. The reviewer feels this paper has some points to be improved and complemented as shown below, before possible publication in Materials.

  1. In Section 2, if the chemical composition was evaluated by XRF, LOI should be contained in Table 1. Furthermore, the authors produced reference samples using OPC. The reviewer feels that it will be better to add the type and chemical composition of the OPC.
  2. In this study, coal gangue and slag were used as binders. Both binders have distinct characteristics according to resources. Therefore, the authors should add more detailed characteristics (such as particle size distribution) of binders.
  3. In line 118, the authors produced reference samples with 0.4 of liquid/solid ratio. However, sand and coarse aggregate can be included in solid when the above word used. Therefore, the authors should exchange to water-to-cement or water-to-binder ratio. Furthermore, is there any reason why the authors used 0.4 of the liquid/solid ratio? It looks seems that reference samples have a different liquid/solid ratio with W- and S-series samples.
  4. In this study, calcined coal gangue powder was used as a binder, and uncalcined and calcined coal gangue were used as a coarse aggregate. Are there any differences between calcined coal gangue powder and calcined coal gangue coarse aggregate except particle size? Furthermore, the reviewer feels that the authors should add the difference of mechanical properties between uncalcined and calcined coal gangue aggregates and the method of calcination of calcined coal gangue aggregates.
  5. In line 176, the authors mentioned that the reason for the enhanced compressive strength of calcined coal gangue concrete compared to the original coal gangue concrete more active SiO2 and Al2O3 produced in calcined coal gangue. However, it cannot be found any basis in the manuscript.
  6. In line 160, the authors explained that Figs. 1 (a) and (b) show the growth rates of ACSC compressive strength at 28 d and 90 d, respectively. However, this explanation corresponds to only Figs. 1 (a). 
  7. In line 186, the authors explained that the reason for a decrease in the compressive strength of samples at 90 days of curing is the internal structure having holes of coal gangue. However, the coal gangue in samples at 28 days of curing had the same internal structure having holes. It is not sufficient for the reason for the compressive strength decreased at 90 days of curing.
  8. In line 198, the authors mentioned that the alkali-activated coal gangue concrete in general has higher compressive strength than the original coal gangue group concrete. Is it the authors’ experimental results? If so, the authors should add more results which can support the authors’ mention. If not, the authors should add some references.
  9. In line 201, the authors explained that the strength of 100% replacement coal gangue increased by 20.25%. However, the authors’ explanation is different from Fig. 2.
  10. There are some grammar and contextual errors in this manuscript. For example,
    1. The sentence in line 110 has two verbs.
    2. In line 115, Table 2 -> Table 3
    3. In Fig. 1 (a), S10 -> S100

Author Response

Dear reviewer:

Many thanks for the comments and suggestions of the paper. The following are the answers and revisions I have made in response to the questions and suggestions. Words in blue are the changes I have made in the text.

  1. In Section 2, if the chemical composition was evaluated by XRF, LOI should be contained in Table 1. Furthermore, the authors produced reference samples using OPC. The reviewer feels that it will be better to add the type and chemical composition of the OPC.

Response: Thank you for your advice and we have revised them. Thank you for your advice and we have added the LOI and the type and chemical composition of the OPC for concrete in the manuscript.

Table 1. Chemical composition of OPC, coal gangue and slag (wt.%).

Precursor

Al2O3

SiO2

CaO

Fe2O3

Na2O

MgO

TiO2

LOI

OPC

4.65

21.18

63.25

3.78

0.18

3.26

1.55

2.15

Slag

14.04

30.58

38.43

0.35

0.57

10.57

1.93

1.17

Coal gangue

36.78

56.56

0.62

1.95

0.42

0.22

2.10

1.32

  1. In this study, coal gangue and slag were used as binders. Both binders have distinct characteristics according to resources. Therefore, the authors should add more detailed characteristics (such as particle size distribution) of binders.

Response: Thank you for your advice and we have revised them. The median sizes d50 of the coal gangue and slag are 17.365  and 10.532, separately.

  1. In line 118, the authors produced reference samples with 0.4 of liquid/solid ratio. However, sand and coarse aggregate can be included in solid when the above word used. Therefore, the authors should exchange to water-to-cement or water-to-binder ratio. Furthermore, is there any reason why the authors used 0.4 of the liquid/solid ratio? It looks seems that reference samples have a different liquid/solid ratio with W- and S-series samples.

Response: Thank you for your advice. The ratio of liquid to solid is 0.4 by water consumption test of standard consistency. We have changed the OPC liquid-solid ratio to water-binder ratio. The OPC water-binder ratio is water/cement. Meanwhile, in geopolymer concrete, Na2SiO3 and NaOH complex solution were employed as activator, and the silica modulus n (molar SiO2/Na2O ratio) was 1.3 (water glass is 65% water).The solid content includes coal gangue, slag and water glass (26.5% SiO2、8.5% Na2O).  

The formula is as follows:

Liquid-solid ratio===0.4

  1. In this study, calcined coal gangue powder was used as a binder, and uncalcined and calcined coal gangue were used as a coarse aggregate. ①Are there any differences between calcined coal gangue powder and calcined coal gangue coarse aggregate except particle size? ② Furthermore, the reviewer feels that the authors should add the difference of mechanical properties between uncalcined and calcined coal gangue aggregates and ③ the method of calcination of calcined coal gangue aggregates.

Response: Thank you for your advice.

①  The difference between calcined coal gangue powder and calcined coal gangue coarse aggregate lies in the different particle size.

② The crushing values of calcined and uncalcined coal gangue have been measured in Table 1, and the mechanical properties of coarse aggregate will be studied next.

③ Fig. 1 shows the production process of coal gangue aggregate, the coal gangue was crushed by a jaw crusher, grinded by a small ball mill and calcined in a muffle furnace at 700℃ for 2 h.

Fig 1.Production process of coal gangue aggregate

  1. In line 176, the authors mentioned that the reason for the enhanced compressive strength of calcined coal gangue concrete compared to the original coal gangue concrete more active SiO2and Al2O3produced in calcined coal gangue. However, it cannot be found any basis in the manuscript.  

Response: Thank you for your advice and we have revised them. Some researchers have discovered that the active SiO2 and Al2O3 components in the coal gangue, with cement hydration products at a certain degree of secondary hydration reaction, can facilitate a more thorough hydration process and improve the microstructure and macro mechanical properties of cement mortar [1,2]. The contents of active Al and Sl in calcined coal gangue at 700℃ were the highest measured by ICP(Inductively Coupled Plasma)[2]. Under the same substitution rate, the strength of calcined gangue concrete is better than that of uncalcined gangue concrete, this is because after the gangue coarse aggregates were calcined, its surface steady-state crystals were cracked and destroyed, and the content of active SiO2 and Al2O3 increased, which endowed the coal gangue coarse aggregates with pozzolanic activity [2,3]. The activated SiO2 and Al2O3 obtained by calcining coal gangue coarse aggregate can also react with the composite alkali-activator to generate C-S-H and other gels. Moreover, the C-S-H gel made the bonding of calcined coal gangue and cementitious materials denser, and the overall concrete structure became more compact. Therefore, the calcined coal gangue geopolymer concrete displayed better compressive properties than the uncalcined coal gangue groups.

[1] Di Wu , Baogui Yang, Yucheng Liu, Pressure drop in loop pipe flow of fresh cemented coal gangue–fly ash slurry: Experiment and simulation, Advanced Powder Technology 26 (2015) 920-922.http://dx.doi.org/10.1016/j.apt.2015.03.009

[2] Z.C. Dong, J. W. Xia, C. Fan, J.C. Cao, Activity of calcined coal gangue fine aggregate and its effect on the mechanical behavior of cement mortar, Construction and Building Materials100 (2015) 63-69.

http://dx.doi.org/10.1016/j.conbuildmat.2015.09.050

[3] H.Q Ma, H.G. Zhu, C. Yi, J.C. Fan, H.Y. Chen, X.N. Xu, T. Wang, Preparation and Reaction Mechanism Characterization of Alkali-activated Coal Gangue–Slag Materials ,Materials 12(2019) 2250. https://doi.org/10.3390/ma12142250

  1. In line 160, the authors explained that Figs. 1 (a) and (b) show the growth rates of ACSC compressive strength at 28 d and 90 d, respectively. However, this explanation corresponds to only Figs. 1 (a).

Response: Thank you for your advice and we have revised them. Figs. 1 (a) and (b) show the growth rates of ACSC compressive strength at 28 d and 90 d respectively. Fig.1 (a) mainly describes the change law of 28d and 90d compressive strength with the increase of coal gangue content. However, through figs.1 (b) more visual under the different replacement rate of coal gangue polymer concrete 28d-90d strength change rate, when the substitution rate is more than 50%, the 90 d compressive strength of gangue concrete is lower than 28d compressive strength.

  1. In line 186, the authors explained that the reason for a decrease in the compressive strength of samples at 90 days of curing is the internal structure having holes of coal gangue. However, the coal gangue in samples at 28 days of curing had the same internal structure having holes. It is not sufficient for the reason for the compressive strength decreased at 90 days of curing.

Response: Thank you for your advice. That's a very good question .The main reason is that coal gangue is a kind of porous material with strong water absorption (seen Table 1.), and there are a lot of water molecules at the interface of cementitious material and coal gangue coarse aggregate, the water evaporates to form the cavity, the bonding surface appears the pore, is disadvantageous to the coal gangue ground polymer concrete compression strength and the durability. When the content of coal gangue increases, the porosity increases, the self-strength is low, and the later strength decreases because of the high water absorption. When the replacement ratio is more than 50%, the compressive strength of 90 d decreases compared with that of 28 d. With the increase of curing age, the water evaporation in the concrete increases, which leads to the increase of porosity, and the concrete interior and the section transition zone become the weak area, which is not conducive to intensity development.

Table1. Characteristics of the coarse aggregates.

Types

Particle size range

/mm

Apparent density

kg/m3

Crushing index value /%

Water absorption/%

Ordinary gravel

4.75-20

2680

16.8

1.51

Calcined coal gangue

4.75-20

2605

13.2

6.10

Uncalcined coal gangue

4.75-20

2678

13.8

4.09

  1. In line 198, the authors mentioned that the alkali-activated coal gangue concrete in general has higher compressive strength than the original coal gangue group concrete. Is it the authors’ experimental results? If so, the authors should add more results which can support the authors’ mention. If not, the authors should add some references.

Response: Thank you for your advice and we have revised them. Very sorry, here for the writing error, the correct way to write is, The calcined coal gangue polymer concrete than uncalcined coal gangue polymer concrete has better compressive strength.

  1. In line 201, the authors explained that the strength of 100% replacement coal gangue increased by 20.25%. However, the authors’ explanation is different from Fig. 2.

Response: Thank you for your advice and we have revised them.

  1. There are some grammar and contextual errors in this manuscript. For example,
  1. The sentence in line 110 has two verbs.
  2. In line 115, Table 2 -> Table 3
  3. In Fig. 1 (a), S10 -> S100

Response: Thank you for your advice and we have revised them.

Please check the attachment!

Reviewer 4 Report

in Eq. (3.4) it is not clear what is a, e, b, N?

Fig. 10 does not describe what is ITZ?

Author Response

Dear reviewer:

Many thanks for the comments and suggestions of the paper. The following are the answers and revisions I have made in response to the questions and suggestions. Words in blue are the changes I have made in the text.

  1. In Eq. (3.4) it is not clear what is a, e, b, N?

Response: Thank you for your advice. In this paper, N is the times of freeze-thaw cycles;   a, b are the undetermined coefficients in the formula; and e is a constant, which is approximately equal to the 2.71828.

  1. Fig. 10 does not describe what is ITZ?

Response: Thank you for your advice and we have revised them. In Fig.10, ITZ stands for interface transition zone between cement paste and aggregate in concrete.  

Please check the attachment!

Round 2

Reviewer 3 Report

This paper investigates the effect of the usage of coal gangue-slag as a binder and coarse aggregates on mechanical properties and durability of alkali-activated coal gangue-slag concrete (ACSC). The water and salt freeze-thaw resistance, compressive strength, chloride permeation, microstructure, performance mechanism, inner freeze-thaw damage distribution, and mechanics models of ACSC were investigated. The reviewer feels this paper has some points to be improved and complemented as shown below, before possible publication in Materials.

  1. 5th comment of Review #3: references relevant to the comment on “This is because the calcined coal gangue produced more active SiO2 and Al2O3, and it could also react with alkali activator to form C-S-H gels, which made the structure denser, leading to the calcined coal gangue exhibiting better compressive strength than the uncalcined coal gangue.” are needed.

  1. The authors explained in the revision the main reason why a decrease in compressive strength at 90 days of curing. However, it does not coincide with the result shown in Fig. 2. Why?

Author Response

Dear reviewer:

Many thanks for the comments and suggestions of the paper. The following are the answers and revisions I have made in response to the questions and suggestions. Words in blue are the changes I have made in the text.

 Thank you for the kind advice. If there is still any question about this paper, please contact us without hesitation.

Sincerely,

Sen Yang

Round 3

Reviewer 3 Report

This paper investigates the effect of the usage of coal gangue-slag as a binder and coarse aggregates on mechanical properties and durability of alkali-activated coal gangue-slag concrete (ACSC). Authors amended paper according to reviewers’ comments and the reviewer feels that this paper is adequate for publication in Materials